# Effect of Different Colored LED Lighting on the Growth and Pigment Content of *Isochrysis zhanjiangensis* under Laboratory Conditions

**Bu Lv** [1,†], **Ziling Liu** [1,†], **Yu Chen** [1], **Shuaiqin Lan** [1], **Jing Mao** [1], **Zhifeng Gu** [1,2], **Aimin Wang** [1,2], **Feng Yu** [1], **Xing Zheng** [1,2,*] and **Hebert Ely Vasquez** [1,2,*]

[1]  Department of Aquaculture, College of Marine Sciences, Hainan University, Haikou 570228, China
[2]  State Key Laboratory of Marine Resource Utilization in South China Sea, Hainan University, 58 Renmin Avenue, Haikou 570228, China
*  Correspondence: zhengxing_edu@163.com (X.Z.); hebertely@163.com (H.E.V.)
†  These authors contributed equally to this work.

**Abstract:** Light is one of the most important environmental factors affecting the growth and reproduction of algae. In this study, the effect of various LED colors on the productivity, chlorophyll (Chl-*a*, Chl-*b*, and total Chl), protein, and carbohydrate content of *Isochrysis zhanjiangensis* in indoor culture was investigated. Microalgae monocultures were cultivated under five different colors (red, green, blue, yellow, and white) for twenty-one days. The microalgae cultured under red light exhibited a higher specific growth rate ($0.4431 \pm 0.0055\ \mu\ \text{day}^{-1}$), and under white light a higher productivity ($0.0728 \pm 0.0013\ \text{g L}^{-1}\ \text{day}^{-1}$). The poorest performance was observed under yellow and green lights. Interestingly, green light exhibited the highest levels of chlorophylls (Chl-*a*, $1.473 \pm 0.037\ \text{mg L}^{-1}$; Chl-*b*, $1.504 \pm 0.001\ \text{mg L}^{-1}$; total Chl, $2.827 \pm 0.083\ \text{mg L}^{-1}$). The highest protein content was observed under the white light ($524.1935 \pm 6.5846\ \text{mg L}^{-1}$), whereas the carbohydrate content was remarkably high under the blue light ($24.4697 \pm 0.0206\ \text{mg L}^{-1}$). This study is important in terms of the selection of light at the appropriate color (wavelength) to increase the content of organic compounds desired to be obtained indoors with the potential for commercially produced cultures.

**Keywords:** *Isochrysis zhanjiangensis*; light-emitting diodes; growth; chlorophyll; organic compounds

## 1. Introduction

The microalgae *Isochrysis zhanjiangensis*, a marine single-cellular golden-brown flagellated species isolated from Nansan Island of Zhanjiang of Guangdong Province, China [1], is an important species in the aquaculture economy and commonly used in the fodder industry and various mariculture systems [2]. Due to the small size, fast-growing, cellulose-free cell walls, and nutrient-richness, especially in polyunsaturated fatty acid (Omega-3), chlorophylls, and carotenoids, *Isochrysis zhanjiangensis* has been mass-produced for feeding fish, shrimp, shellfish seedlings, as well as larvae of a variety of aquaculture animals [3,4]. In particular, *I. zhanjiangensis* is used as a food supply worldwide during broodstock hatchery conditioning. Additionally, it is especially used to culture suspension-feeding larvae and early juvenile bivalve mollusks, because of its nutritional properties that support shell growth and the high survival rates when used as a mono-species diet [5,6].

The growth and propagation of microalgae are affected by several environmental factors, such as temperature, salinity, light, and pH. Light, the main source of energy for algae growth, is one important key factor in regulating its growth and development [7]. Photosynthetic microorganisms do not utilize the whole solar spectrum but only a fraction of it, in particular from 400 to 700 nm. The absorption wavelengths of visible light by algae and plants are mainly concentrated in the blue-violet light region of 400–510 nm and the red-orange light region of 610–720 nm. However, the wavelengths absorbed by microalgae differ

according to species [8]. Several investigations have been focused on either the single or combined influence of light quality (meaning the different wavelengths which are absorbed by water to various extents) [9–12], light quantity (different light intensities) [13–19], and light periodicity (different photoperiods) [20–23], thus indicating that illumination is a complex external factor for microalgae cultivation. Moreover, researchers have found that light quality plays an important role in regulating the growth and development, morphology, photosynthesis, and metabolism of algae [24–27]. For instance, red light is an efficient light quality for the growth of *Arthrospira (Spirulina) platensis*, while it has a significant inhibitory effect on the chlorophyll content [28]. The cultivation of chlorophytes under a mix of green and blue LEDs may prove optimal for growth, biomass productivity, pigments, proteins, and lipids [29–32]. Green light enhanced growth rates, protein, and lipid contents in *Brachiomonas submarina*, and pigment content in *Kirchneriella aperta*. High- and low-intensity green LEDs enhanced lutein biosynthesis compared to red or blue LEDs in *B. submarina* and *Scenedesmus obliquus* [33,34]. High-intensity blue LEDs increased the carotenoid zeaxanthin, and white light was optimal for phycobiliprotein in *Rhodella* sp. and fucoxanthin content for *Stauroneis* sp. and *Phaeothamnion* sp. [33]. Although, the use of any white light sources (fluorescent lamps, RGB LEDs, and white LEDs) for the cultivation of green algae seems to not affect growth. A species-specific response of algae to light intensity has been described in *Desmodesmus quadricauda*, *Parachlorella kessleri*, and *Chlamydomonas reinhardtii* [35]. In *P. kessleri* cells, the concentration of pigments decreased with increasing light intensity, a response found not only in the genus *Chlorella* [13,36–38], but also in other green algae [39].

Light quality, intensity, and photoperiod also affect the growth, biochemical composition, and physiology of *Isochrysis* sp. [40–42]. Microalgal pigments change with algal variety. Therefore, the influences of different light qualities on the physiological properties of algae, such as growth, photosynthesis, and cellular metabolism, are diverse [43]. The ability of the microalgae to utilize different light qualities is determined by this composition of pigments in their cells, and different pigments absorb different light qualities. The growth and development of microalgae and the generation of metabolites are related to light quality, and the light quality that is most suitable for the growth of one microalgae species may not be suitable for another [33]. Therefore, it is of great significance to explore the optimum light quality for the growth of *I. zhangjiangensis*. Thus, this study aims to examine the effects of different LED light qualities on the productivity, chlorophyll, protein, and carbohydrate content of *I. zhanjiangensis* in indoor culture. Our results will aid the optimization of the light conditions for the growth of *I. zhanjiangensis*. Additionally, the results will provide the basis for the optimization of microalgae propagation in indoor conditions and other systems that require artificial illumination in general.

## 2. Materials and Methods

### 2.1. Microalgae Culture Condition

The stock of the microalgae species *I. zhanjiangensis* was obtained from the Microalgae Laboratory of the College of Oceanography at Hainan University (Haikou, China). For the enrichment of the culture media, the nutrient medium Ningbo 3$^{\#}$ was dissolved in filtered and sterilized seawater (29 PSU) with the composition per 1 L of: 100 mg $NaNO_3$, 10 mg $NaH_2PO_4$, 2.5 mg $FeSO_4$, 10 mg EDTA-2Na, 0.25 mg $MnSO_4$, $0.5 \times 10^{-3}$ μg vitamin $B_{12}$, and 6 μg vitamin $B_1$.

The microalgae were cultured in 5 L flat-bottom glass flasks at $25 \pm 1$ °C, with the illumination provided by light-emitting diodes (LED, 191.8 μmoles/m$^2$/s) with a 14:10 h light/dark photoperiod. To enhance growth and prevent the algae from settling, continuous aeration was applied using air stones at 20 L/min. The initial culture media inoculation of the microalgae was 100,000 cells/mL, and the illumination was immediately provided using green (495–530 nm), blue (450–480 nm), red (615–650 nm), white (450–465 nm), and yellow (580–595 nm) light-emitting diodes. Each illumination treatment was set separately to avoid any light interference from the neighboring treatments and set in triplicate.

### 2.2. Measurement of I. Zhanjiangensis Growth

The growth of *I. zhanjiangensis* cultured in the experiment was measured using cell density and biomass (dry weight) to precisely determine the growth pattern of the microalgae. Microalgae cell density was determined daily by counting the cell number using a hemacytometer under a white-light field microscope and was presented as the number of cells/mL. Cell dry weight was estimated by filtering 20 mL of cultured microalgae using a pre-weighed Whatman GF/C filter (47 mm ⌀), washed three times with filtered seawater, and dried to constant weight. Samples were always collected at the same time of the day.

Productivity ($P_x$) and specific growth rate ($\mu$) of *I. zhangjiangensis* were calculated according to the following Formulae [44]:

$$P_x = (X_m - X_i) (T_c)^{-1} \tag{1}$$

$$\mu = (lnX_m - lnX_i) (T_c)^{-1} \tag{2}$$

where, $X_i$ = initial biomass concentration (g/L), $X_m$ = maximum biomass concentration (g/L), and $T_c$ = cultivation time related to the maximum biomass concentration (days).

### 2.3. Measurement of Photosynthetic Pigments, Protein, and Carbohydrate Content

Chlorophyll concentration (Chl) content was determined using a modified method from Jeffrey and Humphrey [45]. Briefly, 20 mL samples were withdrawn from the culture flasks and transferred to opaque plastic bottles, kept away from light, and warmed to room temperature, then filtered using Whatman GF/C filters (25 mm ⌀). After filtration, the filters containing the microalgae biomass were folded and stored separately at $-2\ °C$. Chl extraction was carried out by grinding the filters in 90% acetone (2–4 mL) in a glass homogenizer on an ice bath under low-light conditions for up to 1 min. After grinding, the Chl extracts were transferred to a graduated and stoppered centrifuge tube and rinsed with 10 mL of acetone 90% (10 mL + dead volume of filter). The extract was then centrifuged for 10 min at 500 g. After completion of centrifugation, the absorbance of the supernatant (OD) was measured at 750, 664, 647, and 630 nm against a 90% acetone blank. The concentrations of Chl-*a*, Chl-*b*, and the total Chl were calculated according to the following Equations [45,46]:

$$C_{chla} = (11.85\ E_{664} - 1.54\ E_{647} - 0.08\ E_{630}) \times 10/V \tag{3}$$

$$C_{chlb} = (21.03\ E_{647} - 5.43\ E_{664} - 2.66\ E_{630}) \times 10/V \tag{4}$$

$$C_{tch1} = (20.21\ E_{647} + 8.02\ E_{664}) \times 10/V \tag{5}$$

where, $C_{chla}$, $C_{chlb}$, and $C_{tch1}$ are the concentrations of Chl-*a*, Chl-*b*, and the total Chl (mg/L), and V is the filtered volume (mL).

The protein content of each sample was determined by the Lowry method [47]. Carbohydrates were measured by the phenol sulfuric acid method [48].

### 2.4. Statistical Analysis

Significant differences ($p < 0.05$) among variables were first identified using the *t*-test. Before analysis, data were tested for normality using Kolmogorov–Smirnov's test and for homogeneity of variance using Cochran's C test. All statistical analyses were performed using DPS14.5 software (Hangzhou Rui Feng Information Technology Co., Ltd., Hangzhou, China).

## 3. Results

### 3.1. Specific Growth Rate and Biomass Productivity

The dry weight, specific growth rate, and productivity of *I. zhangjiangensis* under different light conditions are shown in Table 1. The highest and lowest accumulated biomass (dry weight) were observed in the microalgae cultured under the white and red lights, with $1.1761 \pm 0.0212$ and $0.5683 \pm 0.0284$ g/L, respectively. Significant differences



were observed among all the specific growth rate values of *I. zhangjiangensis* cultured under different illumination conditions. The specific growth rates from the highest to the lowest values were under the red, blue, white, and green lights, with $0.4431 \pm 0.0055$, $0.4089 \pm 0.0029$, $0.2948 \pm 0.0011$, and $0.2466 \pm 0.0035$ $\mu$/day, respectively.

**Table 1.** Growth parameters of *I. zhangjiangensis* cultured under different colored LEDs.

| Light Condition | Dry Weight (g/L) | Specific Growth Rate ($\mu$/day) | Productivity (g/L/day) |
|---|---|---|---|
| White | $1.1761 \pm 0.0212$ [a] | $0.2948 \pm 0.0011$ [c] | $0.0728 \pm 0.0013$ [a] |
| Red | $0.5683 \pm 0.0284$ [c] | $0.4431 \pm 0.0055$ [a] | $0.0620 \pm 0.0032$ [b] |
| Blue | $0.6278 \pm 0.0184$ [c] | $0.4089 \pm 0.0029$ [b] | $0.0617 \pm 0.0018$ [b] |
| Green | $0.8928 \pm 0.0572$ [b] | $0.2466 \pm 0.0035$ [d] | $0.0490 \pm 0.0032$ [c] |
| Yellow | $0.7902 \pm 0.0555$ [b] | $0.2158 \pm 0.0034$ [e] | $0.0390 \pm 0.0028$ [d] |

Note: Different lowercase letters represent significant differences between different colored LEDs at the same culture time ($p < 0.05$).

The highest productivity value, a parameter related to potential large-scale microalgae production systems, was observed under the white light ($0.0728 \pm 0.0013$ g/L/day) and was significant when compared to the values observed under the red and blue lights ($0.0620 \pm 0.0032$ and $0.0617 \pm 0.0018$ g/L/day, respectively). Low values were observed under the green and yellow lights, with $0.0490 \pm 0.0032$ and $0.0390 \pm 0.0028$ g/L/day, respectively.

*3.2. Algal Growth*

The daily mean growth of the microalgae *I. zhangjiangensis* cultured under the five different colored LEDs was calculated. The stationary phase was attained after the 11th day from the inoculation in most of the treatments (different light colors), and significant differences were observed when microalgae density values (cells/mL) were compared at their respective stationary phases (Figure 1). The microalgae cultured under white light exhibited a pause in its growing tendency on day 9, but on day 13 resumed exponential cell proliferation until day 18 at the highest value ($1100.0 \times 10^4$ cells/mL). A similar growth pattern was observed in the microalgae cultured under the blue light when cell proliferation paused on day 9 and resumed on day 14, reaching $502.1 \times 10^4$ cells/mL. The microalgae cultured under the green light also exhibited a lag on day 12, but resumed on day 15, reaching maximum cell density, $755.0 \times 10^4$ cells/mL, on day 17. The microalgae that were cultured under the red light reached the stationary phase on day 10 but exhibited the lowest cell density, $0.4 \times 10^7$ cells/mL, among all treatments.

*3.3. Photosynthetic Pigment Production*

Significant differences were observed among the maximum Chl-*a* values at the stationary phase of the different treatments (Figure 2). The maximum Chl-*a* value, $1.473 \pm 0.037$ mg/L, was observed after day 18 of culture under green light. Microalgae cultivated under white light exhibited the second highest value, $1.073 \pm 0.108$ mg/L, on the 21st day. The red light initially exhibited low levels, but after the 9th day of cultivation, its growth accelerated and exhibited the third Chl-*a* value, $0.873 \pm 0.007$ mg/L, on the 11th day. The Chl-*a* values under yellow and blue light were $0.693 \pm 0.008$ and $0.675 \pm 0.002$ mg/L, respectively, and were the lowest levels of Chl-*a* concentration among all the treatments.

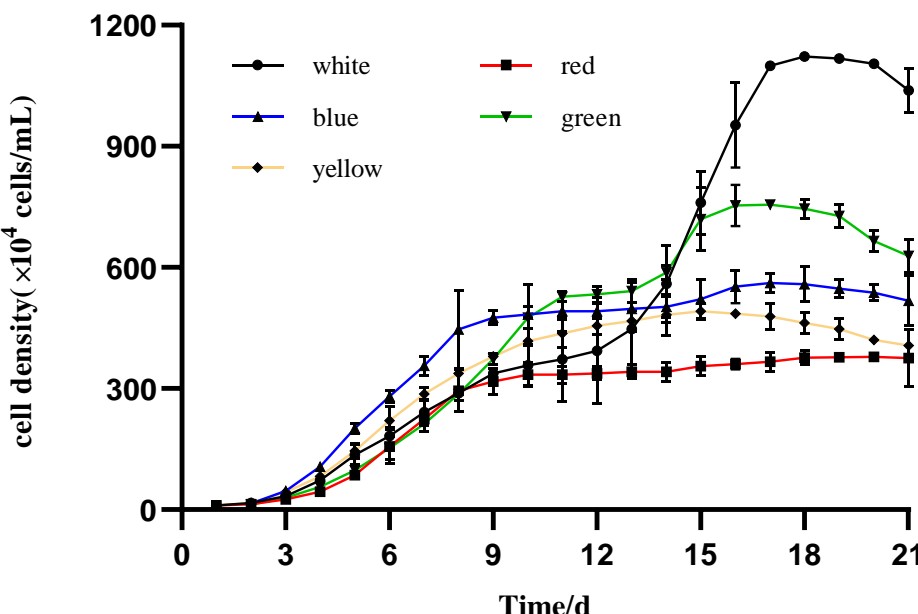

**Figure 1.** Daily mean growth of *I. zhangjiangensis* cultured under different colored LEDs (n = 3, $p < 0.05$ *t*-test).

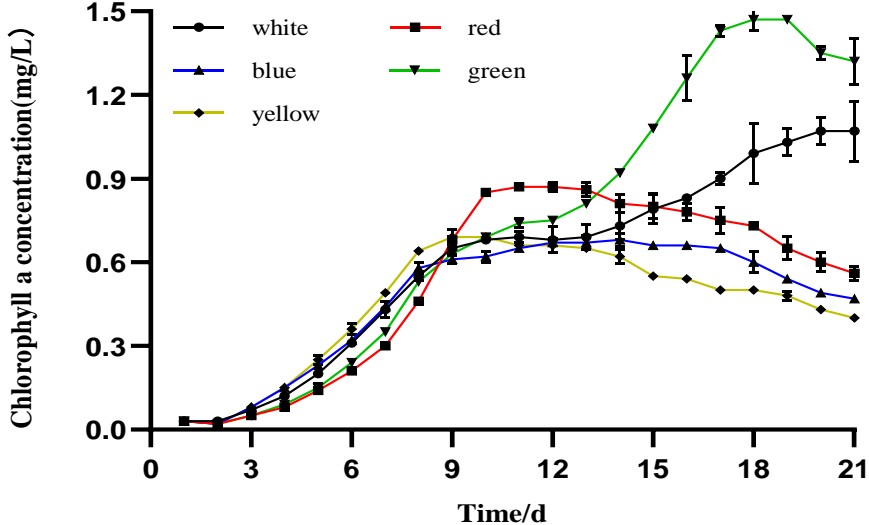

**Figure 2.** Daily mean values of chlorophyll-*a* in *I. zhangjiangensis* cultured under different colored LEDs (n = 3, $p < 0.05$ *t*-test).

The Chl-*b* levels in the five different colored LEDs are shown in Figure 3, and significance was observed at the maximum levels attained among treatments. The maximum Chl-*b* level, 1.504 ± 0.001 mg/L, was observed on the 21st day of culture under the green light. The white light evoked the second highest value (0.918 ± 0.001 mg/L), occurring the same day as the green light. The orange, red, and blue lights evoked lower Chl-*b* levels, with values of 0.360 ± 0.001, 0.275± 0.009, and 0.232 ± 0.003 mg/L, respectively.

The total Chl concentrations of the microalgae *I. zhangjiangensis* cultured under the five different colored LEDs are shown in Figure 4. In the case of the green and white lights, the highest levels were observed on day 21, with values of 2.827 ± 0.083 and 2.238 ± 0.083 mg/L, respectively. Under the red, yellow, and blue lights, the highest mean total values were observed within the 11th and 13th days of cultivation, with corresponding values of 1.095 ± 0.002, 1.000 ± 0.002, and 0.872 ± 0.049 mg/L, respectively.



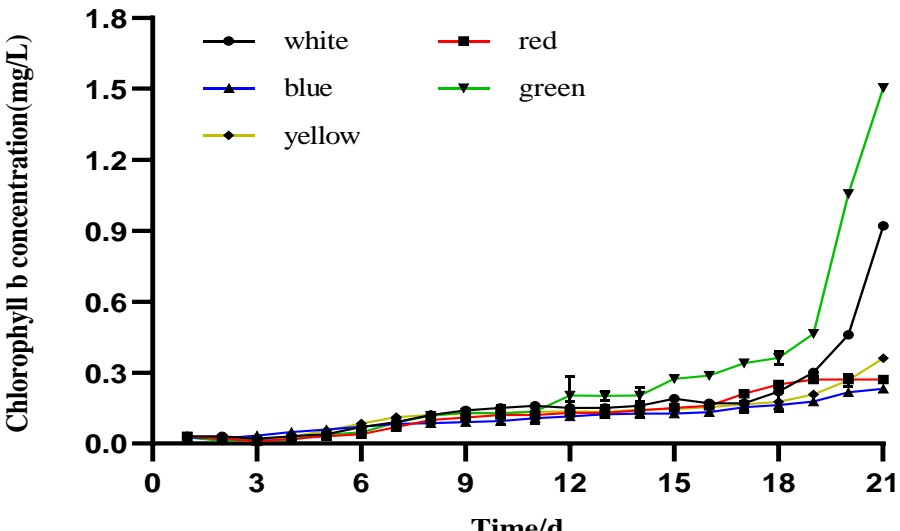

**Figure 3.** Daily mean values of chlorophyll-*b* in *I. zhangjiangensis* cultured under different colored LEDs (n = 3, $p < 0.05$ *t*-test).

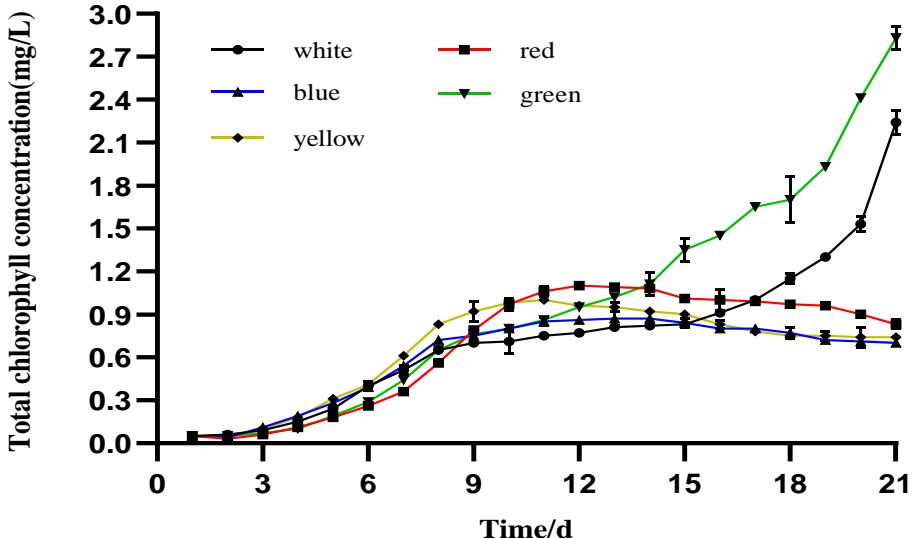

**Figure 4.** Daily mean values of total chlorophyll in *I. zhangjiangensis* cultured under different colored LEDs (n = 3, $p < 0.05$ *t*-test).

### 3.4. Protein and Soluble Carbohydrates' Production

Significant differences were observed in the total protein content of *I. zhangjiangensis* grown under different light colors (Table 2). A particularly high content ($524.1935 \pm 6.5846$ mg/L) was observed in the microalgae cultured under the white light. No significance was observed between the blue and yellow lights, in which both lights exhibited lower content than in the white light. The microalgae cultivated under green and red lights observed the lowest values.

**Table 2.** Protein and carbohydrate content in *I. zhangjiangensis* cultured under different colored LEDs.

| Light Color | Maximum Total Protein Content (mg/L) | Maximum Carbohydrate Content (mg/L) |
|---|---|---|
| White | 524.1935 ± 6.5846 [a] | 8.5859 ± 0.0206 [d] |
| Red | 403.2258 ± 6.5846 [c] | 9.1667 ± 0.0206 [c] |
| Blue | 440.0325 ± 3.3723 [b] | 24.4697 ± 0.0206 [a] |
| Green | 406.6346 ± 3.3548 [c] | 9.3687 ± 0.0206 [b] |
| Yellow | 454.9462 ± 3.4375 [b] | 7.8455 ± 0.1247 [e] |

Note: Different lowercase letters represent significant differences between different colored LEDs ($p < 0.05$).

The mean values of the carbohydrate content were also significant among all light color qualities (Table 2). The highest mean value was 24.4697 ± 0.0206 mg/L, observed in the microalgae cultured under blue light. The green, red, white, and yellow lights exhibited significant mean values within them, varying from 7.8455 ± 0.1247 to 9.3687 ± 0.0206 mg/L.

## 4. Discussion

### 4.1. Specific Growth Rate and Biomass Productivity of I. zhangjiangensis

The present study examined the effect of different colored LED illumination on the productivity, Chl, protein, and carbohydrate content in the indoor culture of microalgae *I. zhangjiangensis*. The highest specific growth rate was observed under the red light, 1.5-fold higher when compared to the white light. Whereas the highest biomass observed under the white light was 2.0-fold higher than that under the red light (Table 1). This observation seems contradictory; however, it is possible that the size of the cells in the microalgae cultured under white light was larger than under the other light colors. Unfortunately, cell size data were not collected at any stage during the experimental period to fully support this idea. Nevertheless, it has been reported that light quality regulates the cell size of the microalgae [49]. For instance, the cell sizes of the green microalgae *C. reinhardtii* grown under blue light were 1.3 and 1.6 times larger than under white and red light sources, respectively, due to a delay in cellular division processes [50]. Additionally, the light intensity has resulted in cell enlargement by a factor of 2.5 in these species, compared to 1.9 in *D. quadricauda*, and only 1.3 in *P. kessleri*. The smaller increase in cell size in *P. kessleri* was compensated for by a 13.6-fold daily increase in cell number under optimal conditions, as compared with a 9.7-fold increase in *C. reinhardtii* and *D. quadricauda* [35]. Another study using different strains of *Chlorella* sp. revealed that the light spectrum had a significant influence on microalgae cell size. The largest and smallest cells were observed under blue and red lights, respectively [51]. Such an increase in cell size is a specific response of organisms that divide by multiple fission and thus can respond to better growth conditions beyond a simple increase in the cell number. The larger cell size is a mechanism that supports better growth in the next cell cycle, possibly leading to better productivity [35]. Nevertheless, it should be emphasized that although the value of the specific growth rate is important, the most relevant parameter related to potential large-scale microalgae production systems is biomass productivity, which was highest when *I. zhangjiangensis* was cultivated under white light. It has been proven that multi-chromatic white light, whether provided by a fluorescent lamp or by LED, is more advantageous than monochromatic light for promoting the growth of *Isochrysis* sp. [42,52,53], which can also be supported by our results. A similar observation has already been reported in the productivity of *Nannochloropsis* sp. cultured under different colored lights, where pink and white lights exhibited higher biomass productivity [54]. These results are probably related to the differences in energy provided by light and captured by the photosynthetic apparatus of the photosynthetic microorganisms [55]. Particularly, between 380 and 750 nm, the energy content is sufficient to produce chemical changes in the absorbing molecules, as happens throughout the photosynthetic pathways prevailing in the microalgae [56].

*4.2. The Effect of Different LED Colors on the Pigment Content of I. zhangjiangensis*

Microalgae, similar to plants, capture light energy (light-harvesting antennas) and produce electrons in the reaction center of the photosystems. For efficient photosynthesis, preserving an excitation balance between the two photosystems (PSI and PSII) is of prime importance. To serve this purpose, microalgae possess specific light-harvesting antennas to expand the available light wavelength. Certain groups of algae contain accessory pigments that help in efficiently harvesting light for photosynthesis [54]. Green algae, in particular, possess a chlorophyll–protein complex which is comprised of Chl-*a* and *b* and carotenoids for carrying out the photosynthesis [57]. Chl molecules absorb light energy and transfer this energy to the photochemical reaction centers presented in algae, cyanobacteria, and higher plants by PSI and PSII, where charge separation occurs. Upon illumination, two electrons are extracted from water, mostly ($O_2$ is evolved), and transferred through a chain of electron carriers to produce one molecule of $NADPH_2$ (nicotinamide adenine dinucleotide hydrogen). Simultaneously, protons are transported from an external space (stroma) into the intra-thylakoid space (lumen), forming a pH gradient. According to Mitchel's chemiosmotic hypothesis, the gradient drives ATP synthesis, which is catalyzed by the protein complex called ATPase or ATP synthase—a reaction called photophosphorylation [58]. In the present investigation, for *I. zhangjiangensis*, the highest Chl content was attained under green light, while blue light resulted in the lowest Chl content, indicating that different light qualities can evoke different levels of photosynthetic pigments. Green light has promoted the production of Chl in *Chlorella vulgaris* [46], and also allowed improvement of *S. platensis* growth [10,59]. A different result has been reported for *I. galbana*. The highest contents of Chl-*a*, Chl-*c*, and Car were obtained under white light, while blue light resulted in the lowest pigment contents [53]. The same report described the highest light absorption in cells cultivated under blue light, but the photochemical reaction was lowered. Cells cultivated under blue and red lights were, respectively, restricted by downregulated photosynthetic efficiency and sufficient light absorption. Meanwhile, green light showed an increase in photosynthetic efficiency, associated with a light absorption close to that for cells exposed to white light, suggesting that green light promotes the photosynthesis of *I. galbana* by balancing the light absorption and utilization [53]. Light with a shorter wavelength, for example, blue light, has a higher probability to cause growth photo-inhibition by striking the light-harvesting complex of cells at its peak electrical energy due to its high energy [60].

Studies have shown that infrared light can cause cell damage [61], while in some multicellular algae, blue light can significantly increase the content of algal photosynthetic pigments, increasing photosynthetic efficiency and ultimately, the growth rate [62,63]. The content of photosynthetic pigments in *I. zhangjiangensis* is the highest under green light, which is conducive to the accumulation of its biomass, while the content of photosynthetic pigments is the lowest under blue light. A previous study reported a contrasting result: low-intensity blue light reduced the pigment content of *Chaetoceros gracilis* but increased it in *I. galbana* [64]. The effect of light qualities on the high-value pigments has also been reported in five microalgae strains from three distinct lineages [33]. In the Rhodophyte *Rhodella* sp., the Chl-*a* levels obtained under red and white LEDs were higher than those reached under green and blue illumination for medium and high intensity. Similarly, the diatom *Stauroneis* sp. also returned a higher Chl-*a* content under medium white light intensities. Contrariwise, in the chlorophyte *K. aperta* and *B. submarina*, the Chl content was significantly higher under blue and green lights at high and medium intensities, returning two-fold higher Chl-*a* compared to red and white LEDs. The cultivation of *Phaeothamnion* sp. under high-intensity blue LEDs also induced a significant increase in Chl-*a*. In general, responses of each strain to different colored LEDs were generally species-specific. These results indicate that the growth performance of different microalgae under different light qualities is not consistent, which may be due to the different compositions of the pigment system of different microalgae, resulting in different requirements for light quality in photosynthesis. It is also known that the content of the photosynthetic pigments increases as light intensity decreases [65]. In the case of *P. kessleri*, the concentrations

of Chl-*a* and *b* and carotenoids decreased with increasing light intensity. Cultures of *C. reinhardtii* and *D. quadricauda* maintained similar levels of photosynthetic pigments at low light intensities, but their concentration increased with the time of cultivation at the highest light intensity [35]. This increase in the photosynthetic pigments is attributed to the need of the microorganisms to improve their photosynthetic efficiency and capture as much energy as possible from light [28]. Considering this, a conclusion can be drawn that when Chl content is high under a specific light color, it does not necessarily imply that this illumination provides adequate amounts of energy for biomass synthesis, and this may be the case of *I. zhangjiangensis* cultured under green light. In addition, light-harvesting ability and energy-transfer processes also differed between green algae species. This has been demonstrated by observing the delayed fluorescence spectra in *C. reinhardtii* and *C. variabilis* cells grown under different light qualities. Both types of green algae primarily modified the associations between light-harvesting chlorophyll protein complexes (LHCs) and photosystems (PSII and PSI) [57].

*4.3. The Effect of Different LED Colors on the Protein and Carbohydrate Content of I. Zhangjiangensis*

The effect of light intensity on microalgal cultures has been extensively studied, and it has been demonstrated that light intensity controls not only the growth rate, but also the lipid storage [17,18,20,21,25,41], structural distribution [66], cellular composition (such as proteins and essential fatty acids) [8], and pigment synthesis [67]. While most research has been focused on the effects of light intensity, it has been shown that light quality also plays a key role in algal metabolism and the effects of light wavelengths on growth are species-specific because of the differences in metabolic pathways, pigmentation, and photoreceptors between species [8]. Moreover, the biochemical composition is a pivotal factor in determining the nutritional value of microalgae. In this investigation, the protein and carbohydrate contents of *I. zhangjiangensis* were influenced by the different LED colors used in the indoor cultures (Table 2). The maximum protein content was measured in the microalgae cultured under white light, followed by the yellow and blue lights, and the lowest level in the red light. Similar results have been reported in microalgal conglomerates of *Chlorella variabilis* and *Scenedesmus obliquus* when the protein content of the microalgal consortia was highest under a cool-white light [68]. Contrasting results have been observed when blue light fluorescent tubes were closely related to protein enhancement in *Isocrhysis* sp. cultured in a bioreactor [42]. However, cell concentration and productivity did not change substantially upon changing the light spectrum during steady-state growth. In addition, experiments conducted with *Tisochrysis lutea* (previously named *I.* aff. *galbana*) under white, blue, green, and red fluorescent lamps in batch cultures revealed that the growth rate and cell density were highest with white light, followed by blue light. Meanwhile, cells under green light had a greater dry weight during exponential growth in comparison with the other light colors, and this monochromatic light also increased the eicosapentaenoic acid and protein contents [69]. In axenic cultures of *Dunaliella tertiolecta* and *Thalassiosira rotula*, blue light also allows higher photosynthetic carbon incorporation into protein than white light [70]. It can be inferred that the optimal light color for the cultivation of *Isochrysis* varies depending on the algal strains and light sources used.

In this study, carbohydrate content was higher in the blue light and the lowest content was in the yellow light. These results concur with the one reported for *A. platensis*, where the highest carbohydrate content was also measured under blue light [28]. However, there are conflicting results on the influence of blue light on microalgae carbohydrate content: In *T. lutea*, it did not change [69], but it decreased in *Isochrysis* sp. [42]. Blue light also enhanced dark respiration, as previously reported for *Scenedesmus obliquus* [71], *Rhodomonas salina* [72], or *D. tertiolecta* and *T. rotula* [70], confirming a higher rate of carbohydrate degradation under blue light. The evidence presented in this study confirms that light quality can affect the biochemical composition of microalgae cells of *I. zhangjiangensis* when they are cultured under different light colors.

## 5. Conclusions

The productivity, Chl (*a*, *b*, and total), protein, and carbohydrate content of *I. zhangjiangensis* can be regulated by different light wavelengths. White light increased the productivity and protein content, and red light increased the specific growth rate. Pigment content was higher under the green light but possibly does not provide adequate amounts of energy for biomass synthesis. The blue light highly promoted carbohydrate content, suggesting that the light influence is a complex phenomenon that is still far from being completely understood. This study is important in terms of the selection of light at the appropriate wavelength to increase the number of metabolites desired in indoor production levels, with the possibility of adaptation to other culture systems that require artificial illumination.

## 6. Ethics Statement

The experiment complied with the regulations and guidelines established by the Animal Care and Use Committee of Hainan University.

**Author Contributions:** B.L., investigation, data curation, visualization, formal analysis, writing—original draft; Z.L., investigation, conceptualization, formal analysis; Y.C., investigation, methodology; S.L., investigation, methodology; J.M., investigation, methodology; Z.G., funding acquisition; A.W., project administration; F.Y., formal analysis; X.Z., conceptualization, methodology, formal analysis, writing—original draft, writing—review and editing; H.E.V., methodology, formal analysis, writing—original draft, writing—review and editing. All authors have read and agreed to the published version of the manuscript.

**Funding:** This work was funded by the Key Research and Development Project of Hainan Province (Grant No. ZDYF2021XDNY277), the Natural Science Foundation for Young Scholars of Hainan Province (Grant No. 320QN207), the Starting Research Fund from Hainan University (Grant No. KYQD-ZR 20061), and the Academician Innovation Center of Hainan Province, China.

**Institutional Review Board Statement:** Not applicable.

**Informed Consent Statement:** Not applicable.

**Conflicts of Interest:** The authors declare no conflict of interest.

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
