# Peer review of "Effect of Different Colored LED Lighting on the Growth and Pigment Content of Isochrysis zhanjiangensis under Laboratory Conditions"

_jmse, doi:10.3390/jmse10111752_

Round 1

Reviewer 1 Report

The manuscript by Bu Lv and Ziling Liu with coworkers concerns the influence of different color of light on the growth parameters of the microalgae Isochrysis zhanjiangensis, as well as on the pigment, protein, and sugar content in such conditions.

I think that in the current view the manuscript is relatively weak and can not be published. My comments are below.

 The idea of the work is not new if don’t take into account the use of a new algae species as a subject. Such works have been performed previously with different microalgae, for example with Chlamydomonas, Desmodesmus, Scenedesmus, Parachlorella (see e.g. [Tim de Mooij et al.,2016; Vitali Bialevich et al., 2022; Rachael M. Morgan-Kiss et al., 2005, as well as Loretta Huang] and others). I’m sure that such works can be found also for cyanobacteria and higher plants. However, in the present manuscript the authors almost don’t cite or discuss the results of those works. I found only a couple of references to works with Spirulina (in Introduction) and Arthrospira (in Discussion). In this concern, the novelty of the work, as well its position among the same works, is not obvious.

The authors don’t describe the algae studied as a species. Is this green, red, or diatom algae, unicellular or colonial one and etc.?

There are mistakes, errors, inaccuracies and missing links through the entire text.

‘The absorption wavelengths of visible light by algae are mainly concentrated in the blue-violet light region of 400-510 nm and the red-orange light region of 610-720 nm’. Is this different in higher plants. I’m sure not.

Reference [7]. The item of the article is ‘Does light have an influence on fish growth?’ In addition, in that article the word ‘algae’ occurs only 1 time in the list of reference (Gulbrandsen, J., Lein, I., Holmefjord, I., 1996. Effects of light administration and algae on first feeding). Thus, this reference is probably incorrect.

‘A large number of studies’ requires few references, which are absent at all.

Reference [11]. Are there more serious work concerning this field?

‘Illumination provided by light emitting diodes (LED, 36-W)’. This is more correctly to indicate the light intensity by mkmol photons/m2s.

‘Carbohydrates were measured by the phenol sulphuric acid method with some modifications’. The modification should be indicated. If the modification was done in the cited work, the authors can not write about modifications in the present study.

The authors don’t explain, including the text of Discussion, why dry weight is the highest at white light, but specific growth rate at this is almost the lowest one. The opposite is at red light, at the same time.

It doesn’t need to indicate P<0.05 in the text. The authors should describe statistics, repetition and ets in the table or figure descriptions.

 ‘However, after day 9th of the culture, its growth accelerated and reached a similar cell density as the one abtained under the white, red, and yellow light’. According to Fig 1 this is not true.

‘Chlorophyll plays an important role in the chloroplast and is mainly responsible for photosynthesis’. It isn’t true. What about proteins of photosystems, ATPase, Rubisco, as well as carotenoids? Chl was abbreviated above.

‘Chlorophyll a play an important role in the growth of organisms as a key pigment’. What about Chl b?

‘has a higher probability to cause photo-inhibition’. Photoinhibition of what?

‘It is also known that’. This requires references.

‘The effect of light intensity on microalgal cultures has been extensively studied, and it has been demonstrated that light intensity controls not only growth rate, but also lipid storage, structural distribution, cellular composition (such as proteins and essential fatty acids), and pigment synthesis.’ This requires references.

‘While the lowest levels of phycobiliproteins have been reported under red light’. Phycobiliproteins are light harvesting complexes in cyanobacteria, red and diatom algae, and high red light induces their reduction, because light intensity is above saturation point. But this is not correctly to compare this with the light induced changers of the total protein content in microalgae. The authors should find another explanation as well as cite works concerning of microalgae. And another question rises here, is the changes in protein content link with photosynthetic apparatus or with something else within an algal cell?

I think that the authors should present microscopic images of algae cells grown at different light colors, including the shape of chloroplast based on auto fluorescence  of Chl. How cell size changes?

Denaturing phoresis and western blot can help to answer a question about changers in the protein content.

PS. It is very inconvenient to work with the text of the manuscript when there is no line numbering.

Author Response

We much appreciate your thorough review and constructive comments on our manuscript. We have carefully considered these comments and replied in a point-by-point manner. The English language has been improved throughout the manuscript.

Reply to Reviewer 1

Comment 1: The idea of the work is not new if don’t take into account the use of a new algae species as a subject. Such works have been performed previously with different microalgae, for example with Chlamydomonas, Desmodesmus, Scenedesmus, Parachlorella (see e.g. [Tim de Mooij et al.,2016; Vitali Bialevich et al., 2022; Rachael M. Morgan-Kiss et al., 2005, as well as Loretta Huang] and others). I’m sure that such works can be found also for cyanobacteria and higher plants. However, in the present manuscript the authors almost don’t cite or discuss the results of those works. I found only a couple of references to works with Spirulina (in Introduction) and Arthrospira (in Discussion). In this concern, the novelty of the work, as well its position among the same works, is not obvious.

Reply 1: Relevant references have been added in the introduction and also results from those investigations have been discussed in the manuscript as suggested by reviewer 1.

Comment 2: The authors don’t describe the algae studied as a species. Is this green, red, or diatom algae, unicellular or colonial one and etc.?

Reply 2: Thanks for your suggestion. The part has been added and reviewed as follows: The microalgae Isochrysis zhanjiangensis, a marine single-cellular golden-brown flagellated species isolated from Nansan Island of Zhanjiang of Guangdong Province, China[1]

Comment 3: The absorption wavelengths of visible light by algae are mainly concentrated in the blue-violet light region of 400-510 nm and the red-orange light region of 610-720 nm’. Is this different in higher plants. I’m sure not.

Reply 3: Thanks for your suggestion. The paragraph has been reviewed as follows: “The absorption wavelengths of visible light by algae and plants are mainly concentrated in the blue-violet light region of 400-510 nm and the red-orange light region of 610-720 nm. However, the wavelengths absorbed by microalgae differ according to species [8].

Comment 4: Reference [7]. The item of the article is ‘Does light have an influence on fish growth?’ In addition, in that article the word ‘algae’ occurs only 1 time in the list of reference (Gulbrandsen, J., Lein, I., Holmefjord, I., 1996. Effects of light administration and algae on first feeding). Thus, this reference is probably incorrect.

Reply 4: Reference has been removed as suggested by the reviewer.

Comment 5: A large number of studies’ requires few references, which are absent at all.

Reply 5: Thanks for your suggestion. The references has been added to the manuscript: “Several investigations have been focused on the either single or combined influence of light quality (meaning the different wavelengths which are absorbed by water to various extents) [9–12], light quantity (different light intensities)[13–19], and light periodicity (different photoperiods) [20–23]

Comment 6: Reference [11]. Are there more serious work concerning this field?

Reply 6: The following references have been added to the manuscript:

 [29] Yoshioka, M.; Yago, T.; Yoshie-Stark, Y.; Arakawa, H.; Morinaga, T. Effect of High Frequency of Intermittent Light on the Growth and Fatty Acid Profile of Isochrysis Galbana. Aquaculture 2012, 338–341, 111–117, doi:10.1016/j.aquaculture.2012.01.005.

[30] Che, C.A.; Kim, S.H.; Hong, H.J.; Kityo, M.K.; Sunwoo, I.Y.; Jeong, G.-T.; Kim, S.-K. Optimization of Light Intensity and Photoperiod for Isochrysis Galbana Culture to Improve the Biomass and Lipid Production Using 14-L Photobioreactors with Mixed Light Emitting Diodes (LEDs) Wavelength under Two-Phase Culture System. Bioresource Technology 2019, 285, 121323, doi:10.1016/j.biortech.2019.121323.

[31] Marchetti, J.; Bougaran, G.; Jauffrais, T.; Lefebvre, S.; Rouxel, C.; Saint-Jean, B.; Lukomska, E.; Robert, R.; Cadoret, J.P. Effects of Blue Light on the Biochemical Composition and Photosynthetic Activity of Isochrysis Sp. (T-Iso). J Appl Phycol 2013, 25, 109–119, doi:10.1007/s10811-012-9844-y.

Comment 7: “Illumination provided by light emitting diodes (LED, 36-W)”. This is more correctly to indicate the light intensity by mkmol photons/m2s.

Reply 7: Thanks for your suggestion. This sentence has been revised as follow: “illumination provided by light emitting diodes (LED, 191.8 moles/m2/s) with a 14:10 h light/dark photoperiod”

Comment 8: Carbohydrates were measured by the phenol sulphuric acid method with some modifications’. The modification should be indicated. If the modification was done in the cited work, the authors can not write about modifications in the present study

Reply 8: Thanks for your suggestion. The text has been corrected as follows: “Carbohydrates were measured by the phenol sulphuric acid method [36].

Comment 9: The authors don’t explain, including the text of Discussion, why dry weight is the highest at white light, but specific growth rate at this is almost the lowest one. The opposite is at red light, at the same time.

Reply 9: Thanks for your suggestion. The following paragraph has been added in the discussion: “The highest specific growth rate was observed under the red light whereas the highest biomass and productivity was observed under the white light (Table. 1). This observation seems contradictory; however, it is possible that size of the cells in the microalgae cultured under white light was larger than under the other light colors. Unfortunately, cell size data was not collected at any stage during the experimental period. Nevertheless, it has been reported that light quality regulates the cell size of the microalgae [37]. For instance, the green microalgae Chlamydomonas reinhardtii grown under blue light was larger than the red source due to a delay in cellular division processes [38]. Another study using different strains of Chlorella sp. revealed that the light spectrum had significant influence on microalgae cell size. The largest and smallest cells were observed under blue and red light, respectively [39]”.

Comment 10: It doesn’t need to indicate P<0.05 in the text. The authors should describe statistics, repetition and ets in the table or figure descriptions.

Reply 10: Accepted. The statistics have been described in the tables and figure descriptions as suggested by the reviewer.

Comment 11: However, after day 9th of the culture, its growth accelerated and reached a similar cell density as the one abtained under the white, red, and yellow light’. According to Fig 1 this is not true.

Reply 11: Thanks for your suggestion. The sentence has been revised as follows: “The microalgae cultured under the green light also exhibited a lag at day 12th, but resumed on day 15th, reaching maximum cell density, 755.0 x 104 cells / mL at day 17th”.

Comment 12: Chlorophyll plays an important role in the chloroplast and is mainly responsible for photosynthesis’. It isn’t true. What about proteins of photosystems, ATPase, Rubisco, as well as carotenoids? Chl was abbreviated above.

Reply 12: Thanks for your suggestion. The sentence has been corrected – the word chlorophyll has been abbreviated – and referenced has also been added as follows: “The chloroplast obtain the necessary energy from the absorption of light by chlorophyll (and to some extent from absorption by other pigments in the chloroplast), and it is responsible for photosynthesis [46]

Comment 13: Chlorophyll a play an important role in the growth of organisms as a key pigment’. What about Chl b?

Reply 13: Text has been added as follows: “Among various forms of Chl, Chl-a plays an important role in the growth of organisms as a key pigment that can convert light energy into chemical energy by donating electrons in the electron transport chain. With two absorption peaks, 430 nm (violet-blue light) and 660 nm (deep red light) [47]. Whereas Chl-b supports assembly and accumulation of light-harvesting complexes. In other words, Chl-b transfers the absorbed light energy to Chl-a for primary photochemistry [48] and the maximum absorption peaks are approximately 435 nm (blue light) and 645 nm (red light) [49].”

Comment 14: “has a higher probability to cause photo-inhibition”. Photoinhibition of what?

Reply 14: Thanks for your question. The sentence has been added into the discussion: “Light with a shorter wavelength, for example, blue light, has a higher probability to cause growth photo-inhibition”

Comment 15: “It is also known that”. This requires references.

Reply 15: Thanks for your suggestion. The reference has been added: [36] Somani, B.L.; Khanade, J.; Sinha, R. A Modified Anthrone-Sulfuric Acid Method for the Determination of Fructose in the Presence of Certain Proteins. Analytical Biochemistry 1987, 167, 327–330, doi:10.1016/0003-2697(87)90172-2.

Comment 16: “The effect of light intensity on microalgal cultures has been extensively studied, and it has been demonstrated that light intensity controls not only growth rate, but also lipid storage, structural distribution, cellular composition (such as proteins and essential fatty acids), and pigment synthesis.” This requires references.

Reply 16: Thanks for your suggestion. This sentence has been revised and References have been added in the manuscript as follows: “The effect of light intensity on microalgal cultures has been extensively studied, and it has been demonstrated that light intensity controls not only growth rate, but also lipid storage [17,18,20,21,25,30], structural distribution [56], cellular composition (such as proteins and essential fatty acids) [8], and pigment synthesis [57].”

[17] Takeshita, T.; Ota, S.; Yamazaki, T.; Hirata, A.; Zachleder, V.; Kawano, S. Starch and Lipid Accumulation in Eight Strains of Six Chlorella Species under Comparatively High Light Intensity and Aeration Culture Conditions. Bioresource Technology 2014, 158, 127–134, doi:10.1016/j.biortech.2014.01.135.

[18] Yeesang, C.; Cheirsilp, B. Effect of Nitrogen, Salt, and Iron Content in the Growth Medium and Light Intensity on Lipid Production by Microalgae Isolated from Freshwater Sources in Thailand. Bioresource Technology 2011, 102, 3034–3040, doi:10.1016/j.biortech.2010.10.013.

[20] Sirisuk, P.; Ra, C.-H.; Jeong, G.-T.; Kim, S.-K. Effects of Wavelength Mixing Ratio and Photoperiod on Microalgal Biomass and Lipid Production in a Two-Phase Culture System Using LED Illumination. Bioresource Technology 2018, 253, 175–181, doi:10.1016/j.biortech.2018.01.020.

[21] Babuskin, S.; Radhakrishnan, K.; Babu, P.A.S.; Sivarajan, M.; Sukumar, M. Effect of Photoperiod, Light Intensity and Carbon Sources on Biomass and Lipid Productivities of Isochrysis Galbana. Biotechnol Lett 2014, 36, 1653–1660, doi:10.1007/s10529-014-1517-2.

[25] Rugnini, L.; Rossi, C.; Antonaroli, S.; Rakaj, A.; Bruno, L. The Influence of Light and Nutrient Starvation on Morphology, Biomass and Lipid Content in Seven Strains of Green Microalgae as a Source of Biodiesel. Microorganisms 2020, 8, 1254, doi:10.3390/microorganisms8081254.

[30] Che, C.A.; Kim, S.H.; Hong, H.J.; Kityo, M.K.; Sunwoo, I.Y.; Jeong, G.-T.; Kim, S.-K. Optimization of Light Intensity and Photoperiod for Isochrysis Galbana Culture to Improve the Biomass and Lipid Production Using 14-L Photobioreactors with Mixed Light Emitting Diodes (LEDs) Wavelength under Two-Phase Culture System. Bioresource Technology 2019, 285, 121323, doi:10.1016/j.biortech.2019.121323.

[56] George, B.; Pancha, I.; Desai, C.; Chokshi, K.; Paliwal, C.; Ghosh, T.; Mishra, S. Effects of Different Media Composition, Light Intensity and Photoperiod on Morphology and Physiology of Freshwater Microalgae Ankistrodesmus Falcatus – A Potential Strain for Bio-Fuel Production. Bioresource Technology 2014, 171, 367–374, doi:10.1016/j.biortech.2014.08.086.

[57] Fu, W.; Guðmundsson, Ó.; Paglia, G.; Herjólfsson, G.; Andrésson, Ó.S.; Palsson, B.Ø.; Brynjólfsson, S. Enhancement of Carotenoid Biosynthesis in the Green Microalga Dunaliella Salina with Light-Emitting Diodes and Adaptive Laboratory Evolution. Appl Microbiol Biotechnol 2013, 97, 2395–2403, doi:10.1007/s00253-012-4502-5.

Comment 17: While the lowest levels of phycobiliproteins have been reported under red light’. Phycobiliproteins are light harvesting complexes in cyanobacteria, red and diatom algae, and high red light induces their reduction, because light intensity is above saturation point. But this is not correctly to compare this with the light induced changers of the total protein content in microalgae. The authors should find another explanation as well as cite works concerning of microalgae. And another question rises here, is the changes in protein content link with photosynthetic apparatus or with something else within an algal cell?

Reply 17: Thanks for your suggestion. The following paragraph has been added to the discussion: “Contrasting results has been observed when blue light fluorescent tubes was closely related to protein enhancement in Isocrhysis sp. cultured in a bioreactor [31]. However, cell concentration and productivity did not change substantially upon changing the light spectrum during steady-state growth. In addition, experiments conducted with Tisochrysis lutea (previously named I. aff. galbana) under white, blue, green and red fluorescent lamps in batch cultures revealed that the growth rate and cell density were highest with white light, followed by blue light. Meanwhile, cells under green light had greater dry weight during exponential growth in comparison with the other light colors, and this mono-chromatic light also increased the eicosapentaenoic acid and protein contents [59]. In axenic cultures of Dunaliella tertiolecta and Thalassiosira rotula, blue light also allows higher photosynthetic carbon incorporation into protein than white light [60]. It can be inferred that the optimal light color for the cultivation of Isochrysis varies depending on the algal strains and light sources used.”

Comment 18: I think that the authors should present microscopic images of algae cells grown at different light colors, including the shape of chloroplast based on auto fluorescence of Chl. How cell size changes?

Reply 18: Thanks for your suggestion. Unfortunately, the authors have only full field images of the microalgae cultures, thus imaging quality concerns would be pointed out if such images are included in the manuscript.

Comment 19: Denaturing phoresis and western blot can help to answer a question about changers in the protein content.

Reply 19: Thanks for your suggestion. The authors agree with the reviewer that such methods may probably explain the differences in the microalgae protein contents. However, the results generated with the Lowry’s method are a reliable way of determine the actual protein content, which was the within the scope of this investigation.

Comment 20: It is very inconvenient to work with the text of the manuscript when there is no line numbering.

Reply 20: The manuscript included line numbers in the original format (Word)

Reviewer 2 Report

Effect of different color LED lighting on the growth and pigment content of Isochrysis zhanjiangensis under laboratory conditions is promising as progression use of LED light. However, for a more fluent and intelligible reading experience. There are various typographical and grammatical problems to deal with as well.

The hypothesis should be clearly elaborated in the introduction part with exact references.

Where the materials are submitted and give the Herbarium number.

What is the exact location of the collection material of Isochrysis zhanjiangensis..?.

Result should be clearly described as per the experimental data obtained.

The productivity, chlorophyll (a, b, and total), protein, and carbohydrate content of I. zhangjiangensis can be regulated by different light wavelengths. In my view this work ois very prelimnary. Author should be providing the LED light and ROS. How LED light can induce Stress and its detoxification via enzymatic regulation. This is most vital and increase the paper quality.

Discussion section is poorly written. Needs to be improved.

This form of manuscript cannot be published prior to revision. Therefore, I recommended it for major revision.

Author Response

We much appreciate your thorough review and constructive comments on our manuscript. We have carefully considered these comments and replied in a point-by-point manner. The English language has been improved throughout the manuscript.

Reply to Reviewer 2

Comment 1: The hypothesis should be clearly elaborated in the introduction part with exact references.

Reply 1: Thank you for suggestion. The pertinent references have been added to the introduction as follows: Light quality, intensity and photoperiod affect the growth, biochemical composition and physiology of Isochrysis sp. [29–31]. However, the effect of light quality on the pigment content and growth of I. zhanjiangensisi is still unclear. Thus, this study aims to examine the effects of different LED light qualities on the productivity, chlorophyll, protein, and carbohydrate content of I. zhanjiangensis in indoor culture

Comment 2: Where the materials are submitted and give the Herbarium number.

Reply 2: The materials were not submitted and neither give herbarium number

Comment 3: What is the exact location of the collection material of Isochrysis zhanjiangensis..?

Reply 3: Thanks for your suggestion. The microalgae samples origin has been described in the manuscript: “The microalgae Isochrysis zhanjiangensis, a marine single-cellular golden-brown flagellated species isolated from Nansan Island of Zhanjiang of Guangdong Province, China [1]”.  “The stock of the microalgae species I. zhanjiangensis was obtained from the Microalgae Laboratory of the College of Oceanography at Hainan University (Haikou, P. R.China)”

Comment 4: Result should be clearly described as per the experimental data obtained.

Reply 4: Thanks for your suggestion. Results section has been improved and inaccuracies has been revised and corrected carefully.

Comment 5: The productivity, chlorophyll (a, b, and total), protein, and carbohydrate content of I. zhangjiangensis can be regulated by different light wavelengths. In my view this work ois very prelimnary. Author should be providing the LED light and ROS. How LED light can induce Stress and its detoxification via enzymatic regulation. This is most vital and increase the paper quality.

Reply 5: Authors agree with the reviewer that more in-depth examination of the effect of LED illumination on the microalgae would increase the manuscript quality; however, considering the current research status of this species, it is also primordial the basic data regarding the light quality and its productivity.

Comment 6: Discussion section is poorly written. Needs to be improved.  

Reply 6: Thank you for suggestion. The discussion has been rewritten, and missing references have been added to the discussion. The following paragraphs have been included in the discussion:

     “The highest specific growth rate was observed under the red light whereas the highest biomass and productivity was observed under the white light (Table. 1). This observation seems contradictory; however, it is possible that size of the cells in the microalgae cultured under white light was larger than under the other light colors. Unfortunately, cell size data was not collected at any stage during the experimental period. Nevertheless, it has been reported that light quality regulates the cell size of the microalgae [37]. For instance, the green microalgae Chlamydomonas reinhardtii grown under blue light was larger than the red source due to a delay in cellular division processes [38]. Another study using different strains of Chlorella sp. revealed that the light spectrum had significant influence on microalgae cell size. The largest and smallest cells were observed under blue and red light, respectively [39]”.

    “The maximum protein content was measured in the microalgae cultured under white light. Followed by the yellow and blue light, and the lowest level in the red light. Similar results have been reported in microalgal conglomerates of Chlorella variabilis and Scenedesmus obliquus when protein content of the microalgal consortia was highest under cool-white light [58]. Contrasting results has been observed when blue light fluorescent tubes was closely related to protein enhancement in Isocrhysis sp. cultured in a bioreactor [31]. However, cell concentration and productivity did not change substantially upon changing the light spectrum during steady-state growth. In addition, experiments conducted with Tisochrysis lutea (previously named I. aff. galbana) under white, blue, green and red fluorescent lamps in batch cultures revealed that the growth rate and cell density were highest with white light, followed by blue light. Meanwhile, cells under green light had greater dry weight during exponential growth in comparison with the other light colors, and this monochromatic light also increased the eicosapentaenoic acid and protein contents [59]. In axenic cultures of Dunaliella tertiolecta and Thalassiosira rotula, blue light also allows higher photosynthetic carbon incorporation into protein than white light [60]. It can be inferred that the optimal light color for the cultivation of Isochrysis varies depending on the algal strains and light sources used.”

Round 2

Reviewer 1 Report

In spite of the fact that I rejected this manuscript by Lv et al. in the first round of revision, I received the answers and a new version of the manuscript (v2). During a reading, I realized that the authors have not changed and rewritten significantly the text. Thus, the present work is still weak from the point of science in my opinion and does not take into account the huge set of available data.

Unfortunately, all this forces me to remain with the same opinion that the present manuscript can not be published in the current view.

Below I will list a few significant comments.

The introduction is still without descriptions of data obtained with other species. For example McGree et al., 2020 (The Chlorophyceae Brachiomonas submarina (Chlamydomonadaceae), Kirchneriella aperta (Selenastraceae), Rhodophyceae Rhodella sp. (Glaucosphaeraceae), diatom Stauroneis sp. (Stauroneidaceae) and the Phaeothamnion sp. (Phaeothamniaceae)), or Bialevich et al., 2022 (Chlamydomonas reinhardtii, Desmodesmus quadricauda, Parachlorella kessleri), and etc. In this concern this paragraph is absolutely incomplete.

In the discussion, the authors explain the discrepancy of the results from highest specific growth and the highest biomass and productivity under different color of light by different cell size. It can be true, of course, but no experimental data were presented supporting this suggestion. I don’t think that this is impossible for the authors to obtain such data. Moreover, differences in cell volume have not been observed previously in Chlamydomonas reinhardtii, Desmodesmus quadricauda, Parachlorella kessleri grown at different intensity and color of light [Bialevich et al., 2022]. These data were not discussed and cited in the present manuscript. In addition, even the results of work by Oldenhof et al., 2004, which was cited, can not explain the two-fold differences in the obtained values by the authors (wight – 1.18/0.57 (2.07); red – 0.29/0.44 (0.66)).

I realized that the authors do not understand electrochemical reactions, which take place on the thylakoid membranes during the primary photosynthetic processes. Chlorophyll is a pigment, it can not donate an electron(s). Chl molecules can only absorb light energy and transfer this energy to the photochemical reaction centers presented in algae, cyanobacteria and higher plants by two photosystems, II and I where charge separation occurs. Electrons for the electron-transport chain are taken in PSII from water mostly.

Low-intensity blue light was not resulted in lowest Chl a content in five different algae (see. McGree et al., 2020, Table 2; also Bialevich et al., 2022, Fig 7). The author do not discuss these publications but should be.

Author Response

We much appreciate your thorough review and constructive comments on our manuscript. We have carefully considered these comments and replied in a point-by-point manner. The English language has been improved throughout the manuscript. 

Comment 1: The introduction is still without descriptions of data obtained with other species. For example McGree et al., 2020 (The Chlorophyceae Brachiomonas submarina (Chlamydomonadaceae), Kirchneriella aperta (Selenastraceae), Rhodophyceae Rhodella sp. (Glaucosphaeraceae), diatom Stauroneis sp. (Stauroneidaceae) and the Phaeothamnion sp. (Phaeothamniaceae)), or Bialevich et al., 2022 (Chlamydomonas reinhardtii, Desmodesmus quadricauda, Parachlorella kessleri), and etc. In this concern this paragraph is absolutely incomplete.

Reply 1: The authors have included descriptions in other species as follows: “The cultivation of chlorophytes under a mix of green and blue LEDs may prove optimal for growth, biomass productivity, pigments, proteins and lipids [29–32]. Green light enhanced growth rates, protein and lipid contents in Brachiomonas submarina and pigment content in Kirchneriella aperta. High- and low-intensity green LEDs enhanced lutein biosynthesis compared to red or blue LEDs in B. submarina, and Scenedesmus obliquus [33,34]. High-intensity blue LEDs increased the carotenoid zeaxanthin, and white light was optimal for phycobiliprotein in Rhodella sp. and fucoxanthin content for Stauroneis sp. and Phaeothamnion sp. [33]. Although the use of any white light sources (fluorescent lamps, RGB LEDs, and white LEDs) for the cultivation of green algae seems to have no effect on growth. An species-specific response of algae to light intensity has been described in Desmodesmus quadricauda, Parachlorella kessleri and Chlamydomonas reinhardtii [35]. In P. kessleri cells, the concentration of pigments decreased with increasing light intensity. A response found not only in the genus Chlorella [13,36–38] but also in other green algae [39]”

Comment 2: In the discussion, the authors explain the discrepancy of the results from highest specific growth and the highest biomass and productivity under different color of light by different cell size. It can be true, of course, but no experimental data were presented supporting this suggestion. I don’t think that this is impossible for the authors to obtain such data. Moreover, differences in cell volume have not been observed previously in Chlamydomonas reinhardtii, Desmodesmus quadricauda, Parachlorella kessleri grown at different intensity and color of light [Bialevich et al., 2022]. These data were not discussed and cited in the present manuscript. In addition, even the results of work by Oldenhof et al., 2004, which was cited, can not explain the two-fold differences in the obtained values by the authors (wight – 1.18/0.57 (2.07); red – 0.29/0.44 (0.66)).

Reply 2: The authors agree with the reviewer that the cell size data would probably support the suggested cause for the discrepancy between the biomass and growth rate. However, in our opinion, this would fall in the area that is beyond the scope of the current research work. Authors therefore decided to discuss the discrepancy by citing more evidence related to the relation of cell size and light quality as follows: “The highest specific growth rate was observed under the red light, 1.5 fold when compared to the white light. Whereas the highest biomass observed under the white light was 2.0 fold that at the red light (Table. 1). This observation seems contradictory; however, it is possible that the size of the cells in the microalgae cultured under white light was larger than under the other light colors. Unfortunately, cell size data was not collected at any stage during the experimental period to fully support this idea. Nevertheless, it has been reported that light quality regulates the cell size of the microalgae [49]. For instance, the size cells of the green microalgae C. reinhardtii grown under blue light were 1.3 and 1.6 times larger than in white and red source respectively due to a delay in cellular division processes [50]. Also light intensity has resulted in cell enlargement by factor of 2.5 in these species, when compared to 1.9 in D. quadricauda, and only 1.3 in P. kessleri. The smaller increase in cell size in P. kessleri was compensated for by a 13.6-fold daily increase in cell number under optimal conditions, as compared with a 9.7-fold increase in C. reinhardtii and D. quadricauda [35]. Another study using different strains of Chlorella sp. revealed that the light spectrum had a significant influence on microalgae cell size. The largest and smallest cells were observed under blue and red light, respectively [51]. Such an increase in cell size is a specific response of organisms that divide by multiple fission and thus can respond to better growth conditions beyond a simple increase in cell number. The larger cell size is a mechanism that supports better growth in the next cell cycle, possibly leading to better productivity [35].”

Comment 3: I realized that the authors do not understand electrochemical reactions, which take place on the thylakoid membranes during the primary photosynthetic processes. Chlorophyll is a pigment, it can not donate an electron(s). Chl molecules can only absorb light energy and transfer this energy to the photochemical reaction centers presented in algae, cyanobacteria and higher plants by two photosystems, II and I where charge separation occurs. Electrons for the electron-transport chain are taken in PSII from water mostly.

Reply 3: The following paragraph concerning the photosynthesis in microalgae has been added in the discussion as follows: “Microalgae, similar to plants, capture light energy (light-harvesting antennas) and produce electrons in the reaction center of the photosystems. For efficient photosynthesis, preserving an excitation balance between the two photosystems (PSI and PSII) is of prime importance. To serve the purpose, microalgae possess specific light-harvesting antennas for expanding the available wavelength of light. Certain groups of algae contain accessory pigments which help in efficiently harvesting light for photosynthesis [54]. Green algae, in particular, possess a chlorophyll–protein complex which is comprised of Chl-a and b and carotenoids for carrying out the photosynthesis [57]. Chl molecules absorb light energy and transfer this energy to the photochemical reaction centers presented in algae, cyanobacteria and higher plants by PSI and PSII where charge separation occurs. Upon illumination, two electrons are extracted from water moslty (O2 is evolved) and transferred through a chain of electron carriers to produce one molecule of NADPH2 (Nicotinamide Adenine Dinucleotide Hydrogen). Simultaneously, protons are transported from an external space (stroma) into the intrathylakoid space (lumen) forming a pH gradient. According to Mitchel’s chemiosmotic hypothesis, the gradient drives ATP synthesis, which is catalysed by the protein complex called ATPase or ATP synthase – a reaction called photophosphorylation [58].”

Comment 4: Low-intensity blue light was not resulted in lowest Chl a content in five different algae (see. McGree et al., 2020, Table 2; also Bialevich et al., 2022, Fig 7). The author do not discuss these publications but should be.

Reply 4: The authors highly appreciate the recommendations made by the reviewer. The suggested references have been added and discussed as follows: “The effect of light qualities on the high value pigments have been also reported in five microalgae strains from three distinct lineages [33]. In the Rhodophyte Rhodella sp., the Chl-a levels obtained under red and white LEDs were higher than those reached under green and blue illumination for medium and high intensity. Similarly, the diatom Stauroneis sp. also returned higher Chl-a content under medium white light intensities. Contrarywise, in the chlorophyte K. aperta and B. submarina, the Chl content was significantly higher under blue and green at high and medium intensities, returning two-fold Chl-a compared to red and white LEDs. The cultivation of Phaeothamnion sp. under high intensity blue LEDs also induced a significant increase in Chl-a. In general, responses of each strain to different color LEDs were generally species-specific. These results indicate that the growth performance of different microalgae under different light qualities is not consistent, which may be due to the different compositions of the pigment system of different microalgae, resulting in different requirements for light quality in photosynthesis. It is also known that the content of the photosynthetic pigments increases as light intensity decreases [65]. In the particular case for P. kessleri, the concentrations of Chl-a and b and carotenoids decreased with increasing light intensity. Cultures of C. reinhardtii and D. quadricauda maintained similar levels of photosynthetic pigments at low light intensities, but their concentration increased with time of cultivation on the highest light intensity [35].”

Reviewer 2 Report

NA

Author Response

We much appreciate your thorough review and constructive comments on our manuscript. 

Round 3

Reviewer 1 Report

I received the manuscript by Lv et al. for the third time after I have decided to reject this work twice due to serious reasons. This is unusual for my experience and surprised me.

Of course, the authors have tried to insert all moments according to the comments from me (and probably other reviewers) and, consequently, the text became much better now. I have to say that these inserts were significant. However, I think that such work has to be done by the authors before submitting for the first time or for the second, specially to the journal from Q1.  

In addition, the authors refuse (I don’t know why) to repeat the experiment and visualize the size of cells grown under different light spectra. It will take about 21 days, but this would allow them to assert a fact, not to assume it based on the published data obtained for other species. Especially since there is data also, according to which the volume of algae cells was not changed under growth at different light spectra, as I have mentioned in a previous review. This is one of the important moments of the present work, which may support the idea of the authors, refute it, or even point out errors that may have occurred during measurements.

Thus, I can not recommend this manuscript for publication.